# Eosinophilic Granulomatosis with Polyangiitis after mRNA-1273 SARS-CoV-2 Vaccine

**DOI:** 10.3390/vaccines11081335

**Published:** 2023-08-07

**Authors:** Lucrezia Mencarelli, Laura Moi, Natacha Dewarrat, Matteo Monti, Lorenzo Alberio, Maxime Ringwald, Karolina Swierdzewska, Antiochos Panagiotis, Camillo Ribi

**Affiliations:** 1Division of Internal Medicine, Department of Medicine, Lausanne University Hospital, Lausanne University, 1011 Lausanne, Switzerland; 2Division of Immunology and Allergology, Institut Central des Hôpitaux du Valais, 1951 Sion, Switzerland; 3Division of Haematology and Central Laboratory of Hematology, Lausanne University Hospital, Lausanne University, 1011 Lausanne, Switzerland; 4Division of Immunology and Allergy, Department of Medicine, Lausanne University Hospital, Lausanne University, 1011 Lausanne, Switzerlandcamillo.ribi@chuv.ch (C.R.); 5Division of Diagnostic Radiology, Department of Radiology, Lausanne University Hospital, Lausanne University, 1011 Lausanne, Switzerland; 6Division of Cardiology, Department of Heart and Vessels, Lausanne University Hospital, Lausanne University, 1011 Lausanne, Switzerland

**Keywords:** eosinophilic granulomatosis with polyangiitis, SARS-CoV-2, mRNA vaccine, immune thrombocytopenia

## Abstract

During one of the worst global health crises, millions of people were vaccinated against SARS-CoV-2. In rare cases, new onset systemic inflammatory diseases were reported with temporal coincidence to the vaccination. We describe a case of severe Eosinophilic Granulomatosis with Polyangiitis (EGPA) in a young asthmatic woman, occurring after a second dose of the mRNA-1273 vaccine. She presented with multisystem EGPA with cardiac and central nervous system involvement, complicated by secondary immune thrombocytopenia (ITP). We review the reported cases of EGPA coinciding with SARS-CoV-2 mRNA vaccination. All potentially vaccine-related EGPA cases reported so far occurred within 14 days from immunization. EGPA is very rare with an incidence of 1:1,000,000 inhabitants, and the number of reported post-vaccination EGPA cases lies within the expected incidence rate for the period. While we cannot prove a causal relationship between the vaccine and EGPA onset, the temporal relationship with the vaccine immune stimulation is intriguing, in a disease occurring almost always in adults with asthma and/or chronic rhinosinusitis and driven by an aberrant Th2 lymphocyte activation with hypereosinophilia; nevertheless, cases of inflammatory diseases (IMIDs) emerging in the context of vaccination remain rare and the benefits of preventing severe COVID presentations with SARS-CoV-2 mRNA vaccines remain unquestionable.

## 1. Introduction

During the SARS-CoV-2 global health crisis, hundreds of millions of people were immunized against SARS-CoV-2 with various vaccines. In Switzerland, approximately 6 million people received at least one dose of mRNA SARS-CoV-2 vaccine by November 2022 [1]. From January 2021 to March 2022, the Swiss Agency for Therapeutic Products (Swissmedic) evaluated 13,388 reports of suspected adverse events to the vaccines, with the mRNA-1273 vaccine accounting for 68% of cases [2]. About 38% of adverse events were classified as serious. Note that the reaction severity was reported by non-health-care and health-care professionals alike, with Swissmedic revising events reported as not serious, when they were supposed to be serious, according to the submitted information. The most frequently reported events included fever, headache, fatigue, and nausea, all of which were considered as tolerable side effects not precluding additional vaccination.

In rare cases, there were reports of immune-mediated inflammatory disease (IMID) flares or even new-onset IMIDs after SARS-CoV-2 vaccines [3,4]. We report here a severe case of Eosinophilic Granulomatosis with Polyangiitis (EGPA), occurring in temporal relationship with SARS-CoV-2 vaccination and complicated by immune thrombocytopenia (ITP). This adds to the few reports of flares and new-onset EGPA after mRNA vaccines reviewed below [5,6,7,8,9].

While these cases remain rare and do not discourage active immunization, it is important to investigate the prevalence and incidence of IMIDs as potential side effects of SARS-CoV-2 vaccination (or even of the disease itself). This issue will undoubtedly require further attention and research in the future.

## 2. Materials and Methods

We describe a severe case of EGPA with multisystem manifestations complicated by ITP after mRNA-1273 SARS-CoV-2 vaccination. During patient hospitalization and ambulatory follow-up after her discharge, the investigations, diagnostic label, and specific treatment options were discussed in multidisciplinary sessions including specialists from internal medicine, immunology, and hematology.

All radiology images shown in this case report were analyzed, described, and prepared with the supervision of expert radiologists and cardiologists. 

Moreover, we performed a detailed literature review to search for similar cases manifesting as flare or new-onset EGPA after mRNA SARS-CoV-2 vaccination. As well as ours, all these reports are detailed and resumed in a dedicated table below.

The patient signed an informed consent and our local ethical committee granted approval for publication.

## 3. Case Presentation

A 46-year-old woman of European descent developed fever, asthenia, and myalgia the day after her second dose of the mRNA-1273 vaccine. She was known for allergic rhinosinusitis and moderate asthma but had no other health issues. The patient never tested positive for SARS-CoV-2 and the first vaccine, a month earlier, was well tolerated. Four days after the initial symptoms, she developed severe watery diarrhea with vomiting and consulted the emergency room. Upon admission, she was hemodynamically stable. Her temperature was 38 °C and her oxygen saturation was normal. The abdomen was tender without guarding. The remaining clinical examination was normal. Blood tests showed leucocytosis 17.6 G/L (reference range: 4–10) with neutrophilia 14 G/L (ref. 1.8–7.5), eosinophilia 1.2 G/L (<0.3), and cholestasis. Eosinophilia increased, peaking at 8.4 G/L (Figure 1). 

Total serum IgE was 1361 kU/L (ref. 5–50 kU/l). IgG 4 levels were in the normal range. An extensive search for parasitic, viral, and bacterial infection was negative, as were anti-neutrophil cytoplasmic antibodies (ANCA). Abdominal MRI showed diffuse inflammatory cholangiopathy (Figure 2). 

Hematologic work-up including bone-marrow biopsy and oncogenomic and molecular analysis (BCR-ABL and FISH analysis for PDGFa, PDFGb, FGFR1, PCM-JAK2) showed no arguments for a clonal hypereosinophilic syndrome. Whole-body 18-FDG PET-CT was inconspicuous. The CT scan showed pulmonary interstitial infiltration and a lung function test confirmed an obstructive pattern. Endoscopy revealed ileum and colon eosinophilic infiltrates. During the work-up, the patient developed left leg dystonia and right extrapyramidal syndrome. A brain MRI revealed multiple ischemic lesions (Figure 3). 

High-sensitive troponins T were elevated [peak at 916 ng/l (ref. <14)] and cardiac MRI showed changes consistent with myocarditis (Figure 4).

The patient received anti-aggregation with acetylsalicylic acid and intravenous methylprednisolone pulses, then switched to oral prednisone, with normalization of the eosinophils count (Figure 1). The day after starting methylprednisolone, she developed severe thrombocytopenia (Figure 1). Platelets decreased from 195 G/L (ref. 150–350) to 20 G/L in 24 h with a nadir at 2 G/L three days later. The work-up for thrombocytopenia was negative (heparin-induced thrombocytopenia, viral serologic panel, disseminated intravascular coagulation, thrombotic microangiopathy, and antiphospholipid antibodies). Venous thrombosis was excluded and anti-PF4 antibodies were negative, which spoke against vaccine-induced immune thrombotic thrombocytopenia [10]. The diagnosis of immune thrombocytopenic purpura secondary to EGPA was retained; however, the platelet count did not increase under high-dose glucocorticoids, intravenous immunoglobulin, and weekly romiplostim. Eventually, the thrombocytopenia subsided after four weekly rituximab courses of 375 mg/m^2^ (Figure 1) and blood count, including eosinophils, remained normal thereafter. 

Upon achieving remission, monthly injections with the anti-IL-5 receptor antagonist benralizumab were added and Prednisone was progressively reduced to 5 mg/day. 

Six months after the diagnosis, the patient still had moderate dystonia impairing her daily activities; two years later, the patient is still receiving benralizumab and is doing fine.

## 4. Discussion

Our patient fulfills the latest ACR/EULAR criteria for EGPA (obstructive airway disease; blood eosinophil count >1 G/L; and predominant extravascular eosinophilic inflammation on biopsy) [11] and had involvement of the bowel, biliary tree, heart, and central nervous system. 

EGPA is an immune-mediated disease that develops in patients with asthma and/or chronic rhinosinusitis. ANCA are found in a minority of patients [12]. EGPA may be separated into two clinical entities: a vasculitic and an eosinophilic phenotype. Patients with vasculitis are more often ANCA-positive, while the eosinophilic subtype more often affects the myocardium and has a worse prognosis [12]; there is considerable overlap between the two entities. 

In our ANCA-negative patient, the multiple ischemic cerebral infarctions seen on the MRI are consistent with a small–medium-size vasculitis. It is likely that the high eosinophil count contributed to myocarditis and cerebral ischemia due to the cytotoxic effects on cardiomyocytes and endothelial cells as well as pro-thrombotic effects [13]. 

The association of thrombocytopenia and EGPA is uncommon. There are reports of thrombotic microangiopathies complicating EGPA [14] but this was unlikely in our patient, in the absence of hemolysis and schistocytes. 

An important diagnostic hypothesis to exclude rapidly after vaccination was vaccine-induced thrombosis and thrombocytopenia (VITT). While the majority of the cases of VITT are related to adenovirus-based SARS-CoV-2 vaccines [10,15,16], two possible cases secondary to the mRNA-1273 vaccine have been reported [17,18]; however, most cases of VITT were characterized by venous thrombosis and high-titer anti-PF4 antibodies [10], which was not the case in our patient.

The five-factor score, a useful tool to evaluate the prognosis and guide the intensity of treatment in systemic vasculitis [19], was worryingly high at 2, indicating the need for a second immunosuppressant in addition to systemic corticosteroids.

Given the concomitant presence of refractory ITP, the choice of rituximab over cyclophosphamide was made. Rituximab, associated with corticoids as a second-line treatment or as an induction therapy in ITP, is reported efficacious [20]; moreover, rituximab has shown efficacy in retrospective studies among patients with EGPA refractory to standard therapy [21]. A controlled trial to evaluate rituximab as an induction treatment in EGPA showed no significant difference in achieving remission comparing to standard treatment [22].

EGPA is a rare disease with an estimated prevalence of 10.7 per 14 million adults [12]. In Switzerland, 6 million people have received at least one dose of mRNA vaccine [1] and so far four cases of EGPA after mRNA vaccination have been reported by the Swiss Agency for Therapeutic Products. 

The aim of vaccination is to produce a strong neutralizing immune reaction lasting for years, with few and acceptable side effects; therefore, having a predominant Th1 with a low-Th2 response after inoculation is needed. A Th2-pathway-mediated disease is anecdotally associated with vaccines and allergen-specific immunization procedures [23]. To the best of our knowledge, there is no study on the Th2 response after immunization with SARS-CoV-2 vaccines. A mRNA-1273 vaccine phase I clinical trial has shown a predominant Th1 response after in vitro spike protein stimulation of T cells collected from vaccinated participants [24]. We can hypothesize that in our patient, with her atopic predisposition, asthma, and possibly pre-existing immune dysregulation, repeated administration of the mRNA induced an excessive Th2 response, leading to EGPA. 

We compiled other reports of new-onset EGPA or relapses after prolonged EGPA remission secondary to immunization with mRNA vaccines (Table 1). 

All but one case occurred in patients with pre-existing asthma and or rhinosinusitis. All patients developed symptoms within 14 days from the first or second vaccination. The most common complications were neuropathies and myocarditis. Peak peripheral blood eosinophil values ranged from 0.88 to 13.4 G/L. Half of the cases presented with positive ANCA and all of them were p-ANCA, typically more frequent in EGPA [12]. All needed treatment with immunosuppressants with corticosteroids. Like our patient, another received rituximab while most of the other cases were treated with cyclophosphamide.

In addition to ours, three other cases of EGPA after the mRNA vaccine have been reported to Swissmedic by November 2022, all aged 60 or more. We do not have more details concerning these patients. While we ignore how many spontaneous EGPA cases occurred in the same period, the incidence of the four potentially vaccine-related EGPA cases in a population of eight million people corresponds to the reported natural occurrence of the disease. As with EGPA, other cases of IMIDs were reported to flare or to newly occur after immunization with mRNA vaccines [3,4]. As in our case, while the temporal relationship to the immunization is evident, it is impossible to prove a causal relationship between the vaccine and the IMID. 

## 5. Conclusions

In the near future, it will be interesting to investigate whether the prevalence and incidence of autoimmune diseases have increased secondary to SARS-CoV-2 vaccinations and, if possible, compare them to IMIDs secondary to SARS-CoV-2 infection. Gentiloni and colleagues presented new-onset and flares of IMIDs after SARS-CoV-2 vaccination and disease in a single-center retrospective study but no incidence has been formally analyzed [27]. 

Another interesting point to address in the future would be the stimulation of Th2 lymphocytes caused by SARS-CoV-2 vaccination in atopic patients like ours. 

Up to the present day, these cases remain fortunately rare and do not preclude to use mRNA SARS-CoV2 vaccines, given the established benefits in protecting from severe forms of COVID-19 disease [2]. Finally, even in patients already known for IMID, booster vaccinations are safe and still recommended to elicit a stronger immunization, particularly in patients under immunosuppressant treatments [28].

## Figures and Tables

**Figure 1 vaccines-11-01335-f001:**
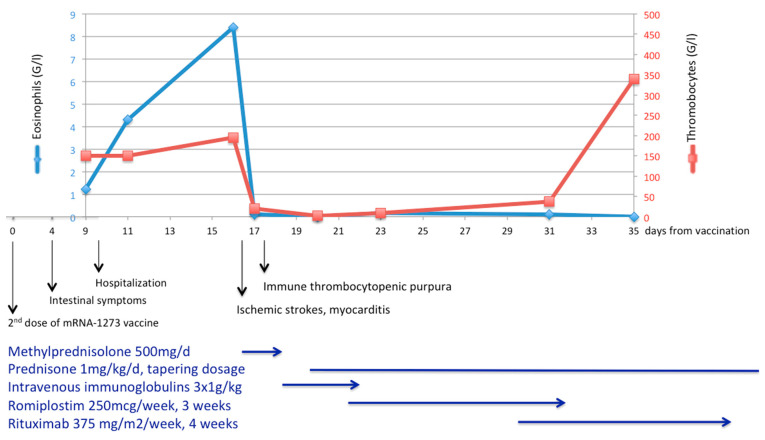
Disease course in a 46-year-old patient with new-onset eosinophilic granulomatosis with polyangiitis occurring after SARS-CoV-2 mRNA 1273 vaccine. Disease manifestations with evolution of blood eosinophil and thrombocyte count and treatment received. Multiple ischemic strokes occurred at the peak of peripheral blood eosinophilia (8.4 G/L).

**Figure 2 vaccines-11-01335-f002:**
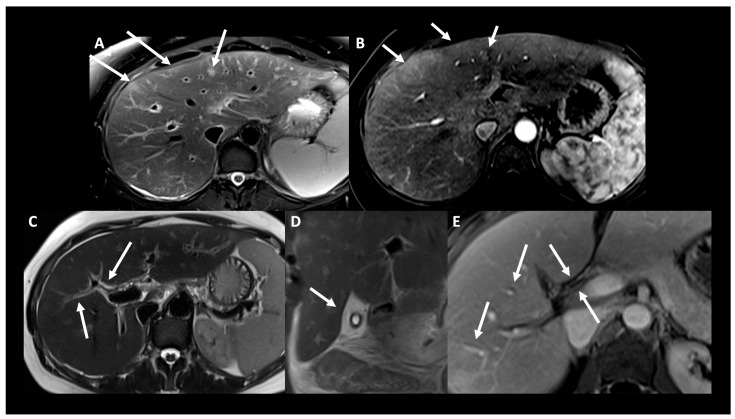
Liver MRI showing cholangiopathy disease. Magnetic Resonance Imaging with WI-weighted images, performed 10 days after symptom onset, showing patchy and ill-defined T2 hypersignal lesions ((**A**), arrow) with discrete arterial phase enhancement ((**B**), arrow). Periportal edema ((**C**), arrow), diffuse gallbladder wall thickening ((**D**), arrow), and biliary wall enhancement on portal venous phase ((**E**), arrows) are seen on T2.

**Figure 3 vaccines-11-01335-f003:**
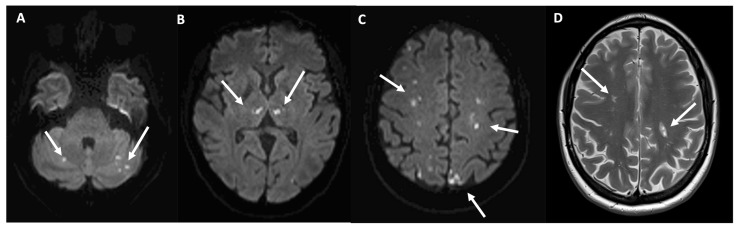
Cranial MRI showing multiple ischemic lesions of the brain. Brain MRI performed 16 days after symptom onset (**A**–**C**). Diffusion-weighted images denote multiple focal acute infarcts in the posterior and anterior circulation territories: in both cerebellar hemispheres ((**A**), arrows), in the thalami ((**B**), arrows) and bilateral cortical and subcortical fronto-parieto-occipital regions ((**C**), arrows). (**D**) Axial T2 weighted MRI performed after 3 months shows residual infarction lesions in corona radiata (arrows).

**Figure 4 vaccines-11-01335-f004:**
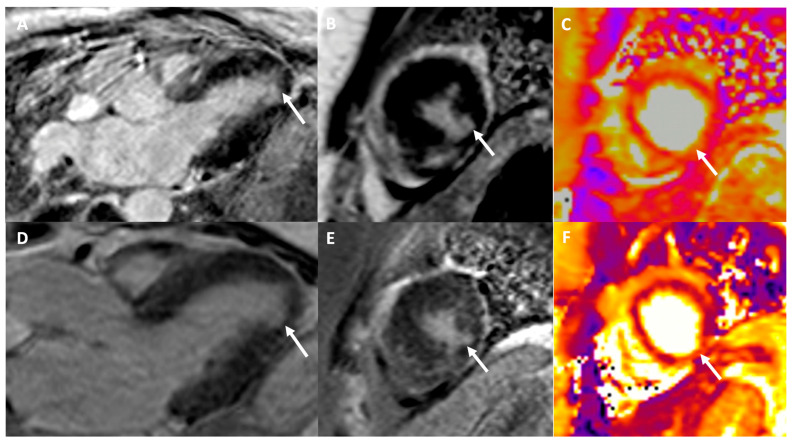
Cardiac MRI consistent with myocarditis. Cardiac MRI performed initially (**A**–**C**) and after 3 months (**D**–**F**). Three chambers (**A**) and short axis (**B**) LGE images demonstrating a focal medio-mural lesion in the inferior and inferolateral apical myocardial segment (arrows), persistent on control MRI 3 months later ((**D**,**E**), arrows). T2-imaging short-axis apical images (**C**,**E**) show a focal T2 hypersignal in the same territory, translating into a myocardial inflammatory edema. LGE: late gadolinium enhancement.

**Table 1 vaccines-11-01335-t001:** Reported cases of new-onset eosinophilic granulomatosis with polyangiitis or disease flares after immunization with mRNA vaccines.

Case	Age, Gender	Pre-Existing Respiratory Disease	Pre-Existing EGPA (Treatment)	mRNA Vaccine Type	Time between Immunization and EGPA Symptom Onset in Days (Vaccine Dose)	Clinical Presentation	Complications	Peak Blood Eosinophil Count (G/L)	ANCA Positivity (Type)	FFS	Treatment Received
[5]	62, F	Asthma	No	BNT161b2	Few days (2nd dose)	Fever, numbness in feet and palms, walking disability, purpura, peri-orbital edema	Polyneuropathy, optic nerves inflammation, myocarditis, vasculitis at cutaneous biopsy	13.4	Yes (anti-MPO)	1	MP, RTX
[6]	63, M	Allergic rhino-sinusitis and asthma	No	mRNA-1273	1 (1st dose)	Diplopia, headache, dry cough	3rd cranial nerve palsy, myopericarditis	12.4	No	2	MP, P, CYC
[9]	79, F	No	No	BNT162b2	14 (2nd dose)	Myalgia, Weakness	Rhabdomyolysis, acute kidney injury, pauci-immune crescentic glomerulonephritis, and interstitial nephritis with prominent eosinophilia	5.3	Yes (p-ANCA, anti-MPO)	1	MP, P, CYC
[7]	79, F	Asthma	No	mRNA-1273	14 (2nd dose)	Progressive paraesthesia of hands and feet, myalgia, walking disability	DVT, vasculitic neuropathy with eosinophil infiltration	12.4	Yes (p-ANCA, anti-MPO)	0	P, AZA
[25]	N/A, M	Rhino-sinusitis and asthma	Yes (untreated for 7 years)	BNT162b2	5 (1st dose)	Weakness, neuropathic pain, blurry vision	Polyneuropathy, mild left foot drop	0.94	No	0	P
[26]	71, F	Rhino-sinusitis and asthma	Yes (MEP, ICS)	BNT162b2	10 (1st dose)	Myalgia, arthralgia, dyspnoea, cough, chest pain, paraesthesia	Mononeuritis multiplex	4.3	Yes (p-ANCA)	0	MP, P, BEN
[8]	63, F	Allergic rhinitis and asthma	Yes (AZA)	mRNA-1273	14 (1st dose)	Dyspnoea, chest pain	Myocarditis	0.88	No	1	MP. P. CYC
[8]	64, M	Asthma and rhinosinusitis with nasal polyps	No (but constitutional symptoms for 3 months)	BNT162b2	2 (1st dose)	Numbness in thigh, purpura, chest pain	Endomyocarditis, cutaneous vasculitis, mononeuritis multiplex, possible glomerulonephritis	5.39	No	1	MP, P, CYC
Present case	46, F	Allergic rhino-sinusitis and asthma	No	mRNA-1273	4 (2nd dose)	Enteritis, fever, fatigue, myalgia	Myocarditis, multiple strokes, immune thrombocytopenia	8.4	No	2	MP, P, IVIG, RPL, RTX, BEN

AZA: azathioprine; FFS: five-factor score; MP: methylprednisolone pulses; P: prednisone; IVIG: intravenous immunoglobulins; RPL: romiplostim; CYC: cyclophosphamide; RTX: rituximab; BEN: benralizumab, DVT: deep venous thrombosis.

## Data Availability

All data is available upon request to the corresponding author.

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
