# Peer review of "Eosinophilic Granulomatosis with Polyangiitis after mRNA-1273 SARS-CoV-2 Vaccine"

_vaccines, 2023, doi:10.3390/vaccines11081335_

Round 1

Reviewer 1 Report

Interesting case report and review of the literature. Appropriately conservative regarding the role of the vaccine in triggering this syndrome.

Author Response

Dear Reviewer,

Thank you for your work and your comments. 

As asked from the editor, in the revised version we added more details about our materials and methods and results, the literature background and possible future prospects reserach.

All new modifications and references are highlighted in the revised manuscript.

Best regards,

Lucrezia Mencarelli et al.

Reviewer 2 Report

The paper is a related case report of severe Eosinophilic Granulomatosis with Pol-21 yangiitis (EGPA) in a young asthmatic woman, occurring after a 2nd dose of mRNA-1273 vaccine.

The case report is very well documented and comprehensive. A rare complication is described and discussed in relation to cases already described in the literature.

The paper deserves due attention for how it was clinically developed by the authors.

Minor editing of English language required.

Author Response

Dear Reviewer,

Thank you for your work and your comments. 

As asked from the editor, in the revised version we added more details about our materials and methods and results, the literature background and possible future prospects research.

All new modifications and references are highlighted in the revised manuscript.

Concerning the minor editing of English language, a native English speaker reviewed the paper.

Best regards,

Lucrezia Mencarelli et al.

Reviewer 3 Report

This is an interesting report describing the case of a sudden eosinophilic granulomatosis with polyangiitis few days after the second immunization with  mRNA-1273 SARS-CoV-2 vaccine, with central nervous and cardial involvement and accompanied with thrombocytopenia. Although mRNA vaccination is more characterized by a Th1 activation, the Th2 overactivation in this case is plausible with regard to the allergic background of the patient. The overall picture from the symptoms and measured labaratory parameters is well presented and discussed in relation to published work. The description of such rare post-vaccination events is of general interest to the readers. The case report can be recommended for publication.

Author Response

Dear Reviewer,

Thank you for your work and your comments. 

As asked from the editor, in the revised version we added more details about our materials and methods and results, the literature background and possible future prospects research.

All new modifications and references are highlighted in the revised manuscript.

Best regards,

Lucrezia Mencarelli et al.